# Acute stress during witnessing injustice shifts third-party interventions from punishing the perpetrator to helping the victim

Huagen Wang[1,2,3], Xiaoyan Wu[1,2,3], Jiahua Xu[4], Ruida Zhu[5], Sihui Zhang[1,2,3], Zhenhua Xu[1,2,3], Xiaoqin Mai[6], Shaozheng Qin[1,2,3,7]*, Chao Liu[1,2,3]*

**1** State Key Laboratory of Cognitive Neuroscience and Learning & IDG/McGovern Institute for Brain Research, Beijing Normal University, Beijing, China, **2** Beijing Key Laboratory of Brain Imaging and Connectomics, Beijing Normal University, Beijing, China, **3** Center for Collaboration and Innovation in Brain and Learning Sciences, Beijing Normal University, Beijing, China, **4** Psychiatry Research Center, Beijing Huilongguan Hospital, Peking University Huilonguan Clinical Medical School, Beijing, China, **5** Department of Psychology, Sun Yat-sen University, Guangzhou, China, **6** Department of Psychology, Renmin University of China, Beijing, China, **7** Chinese Institute for Brain Research, Beijing, China

* szqin@bnu.edu.cn (SQ); liuchao@bnu.edu.cn (CL)

**Data Availability Statement:** The dataset and code for this manuscript are accessible on the Open Science Framework (OSF) at https://osf.io/fkae9/?

## Abstract

People tend to intervene in others' injustices by either punishing the transgressor or helping the victim. Injustice events often occur under stressful circumstances. However, how acute stress affects a third party's intervention in injustice events remains open. Here, we show a stress-induced shift in third parties' willingness to engage in help instead of punishment by acting on emotional salience and central-executive and theory-of-mind networks. Acute stress decreased the third party's willingness to punish the violator and the severity of the punishment and increased their willingness to help the victim. Computational modeling revealed a shift in preference of justice recovery from punishment the offender toward help the victim under stress. This finding is consistent with the increased dorsolateral prefrontal engagement observed with higher amygdala activity and greater connectivity with the ventromedial prefrontal cortex in the stress group. A brain connectivity theory-of-mind network predicted stress-induced justice recovery in punishment. Our findings suggest a neurocomputational mechanism of how acute stress reshapes third parties' decisions by reallocating neural resources in emotional, executive, and mentalizing networks to inhibit punishment bias and decrease punishment severity.

## Introduction

Humans are often willing to interfere in the injustices of others by either punishing the transgressor or helping the victim, even if they are not involved in the unfair event [1]. This extraordinary feature of human society allows us to establish, broadcast, and enforce social norms [2,3]. People typically prefer to punish the offender rather than help the victim in justice restoration [4,5]. Such preferences, however, are often context-dependent [6]. Unpredictable and uncontrollable events are ubiquitous in daily life, and people frequently respond to norm

view_only=
96987a8bd5bf489c98a65c1810342501.

**Funding:** The research was supported by the Scientific and Technological Innovation (STI) 2030-Major Projects 2021ZD0200500 (https://en.most.gov.cn/) , the National Natural Science Foundation of China (https://www.nsfc.gov.cn/english/site_1/index.html, 32130045 to SQ and 32271092 to CL), the Major Project of National Social Science Foundation (http://www.nopss.gov.cn/GB/219469/431028/, 19ZDA363 to CL and 20&ZD153 to SQ), and Beijing Municipal Science and Technology Commission (https://kw.beijing.gov.cn/, Z151100003915122 to CL) , and the Fundamental Research Funds for the Central Universities to SQ. The funders had no role in study design, data collection and analysis, decision to publish, or preparation of the manuscript.

**Competing interests:** The authors have declared that no competing interests exist.

**Abbreviations:** ACC, anterior cingulate cortex; AI, anterior insula; AIC, Akaike information criterion; BOLD, blood oxygen level-dependent; CPT, cold pressor test; DLPFC, dorsolateral prefrontal cortex; EPI, echo-planar imaging; GLM, general linear model; MNI, Montreal Neurological Institute; MU, monetary unit; PPI, psychophysiology interaction; rDLPFC, right dorsolateral prefrontal cortex; ROI, region of interest; SVC, small-volume correction; ToM, Theory of Mind; TPI, third-party intervention; TPIG, third-party intervention game; TPJ, temporoparietal junction; vmPFC, ventromedial prefrontal cortex.

violations under stressful situations, which is a reality that deserves special attention. Decision-making under stress can follow a "tend and befriend" pattern, where decision behavior can be both adaptive and altruistic, promoting survival and well-being at both individual and group levels [7]. Nevertheless, it is important to note that the "tend and befriend" pattern does not always hold true, as some studies have demonstrated a decrease in prosocial preferences under acute stress [8–10], while others have found no significant effects in economic games (according to a recent meta-analysis) [11]. Stress can profoundly impact psychological, neurophysiological, and brain networks involved in third-party decision-making, thereby influencing the affective response, motivation, and subjective value assigned to the decisions [12,13].

The dual process theory of human decision-making emphasizes 2 distinct systems: system 1 is intuitive, fast, and regulated mainly by the limbic system, including the amygdala, whereas system 2 is reflective, slow, deliberate, and regulated mainly by the prefrontal cortex [14,15]. Consistently, according to the biphasic-reciprocal model, stress-related hormones and neurotransmitters increase activity in the emotional salience network at the expense of the executive control network under acute stress [16]. Considerable evidence has shown that exposure to uncontrollable stress can shift brain function [61] from a thoughtful, reflective mode toward a more rapid reflexive response regulated by higher-order prefrontal and primitive neural circuits, respectively [12,17]. Such stress-sensitive neural systems, at least in part, overlap with functional brain systems and networks involved in the decision-making process during third-party responses. Additionally, acute stress has been found to increase activation in brain regions associated with Theory of Mind (ToM), which plays a crucial role in altruistic behavior [18,19].

Third-party punishment and help are both examples of altruistic behavior, and they share a common neuronal mechanism. However, they also involve distinct processes [5]. When facing injustice, both third-party responses correlate with the activation of fairness violations, e.g., the anterior insula (AI), anterior cingulate cortex (ACC), dorsolateral prefrontal cortex (DLPFC), striatal reward circuitry, and emotional regulation systems, including the amygdala and ventromedial prefrontal cortex (vmPFC) [5,20]. Enhanced ventral striatal activity is further associated with punishing an offender instead of helping a victim [5]. On the other hand, the mentalizing network, including the temporoparietal junction (TPJ), precuneus, and mid-temporal regions, is more strongly associated with an individual's preference for helping rather than punishing [21]. In other circumstances, activity in participants' right dorsolateral prefrontal cortex (rDLPFC) increased more after experiencing advantageous unfairness and engaging in help instead of punishment, whereas those who punish more after experiencing disadvantageous unfairness demonstrated significantly decreased activity in the rDLPFC [20]. It is important to note that the DLPFC is not only involved in mentalizing [19] but also plays a critical role in context-dependent cognitive control [22]. Although studies have demonstrated that stressed individuals have greater altruistic [23,24], cooperative, and other-oriented tendencies [25], the neurocognitive mechanisms underlying the trade-off between third-party help and punishment under acute stress remain unclear.

In addition to the trade-off of third-party help and punishment at a behavioral level, it remains unclear whether the 2 candidate decisions involve differential neurocomputational processes; for example, whether and with what severity to intervene in an injustice event under uncontrollable stress [5,26]. The computational model allows us to decipher signals that are not expressed by the behavioral indicators, e.g., in value utility calculations, people rely only on intuition and habit in the allocation of some choices (e.g., helping), while others (e.g., punishing) are considered more carefully and deliberately. We can identify specific neural circuits linked to specific decision variables and processes by combining economic computational tools and neuroimaging approaches. Therefore, it is critical to investigate the neurocomputational bases of third-party intervention under acute stress by integrating

economic paradigms, computational tools, and neuroimaging approaches. Based on the biphasic-reciprocal model of the stress-induced transition from deliberate to intuitive under stress, we hypothesized that people under stress would prefer helping when third-party punishment and helping are in conflict because helping victims is considered a more habitual response in stressful situations, which aligns with "tend and befriend" concept due to its social approach property [7,27]. However, previous research has demonstrated that third-party punishment involves the activation of the DLPFC, which plays a crucial role in executive functioning and the integration of multiple information streams for appropriate decision-making [28–30]. We therefore expected that acute stress would decrease behaviors that require adequate cognitive control while increasing behaviors that recruit habitual and reflexive regulation.

To test the above hypotheses, we set up an event-related fMRI study in conjunction with an economic paradigm of third-party intervention and computational modeling approach to examine the behavioral and neurocomputational bases of third-party interventions under acute stress. A classic cold pressor test (CPT) was used to induce acute stress (**Fig 1A**), and a novel third-party intervention game (TPIG) (**Fig 1E &1F**) was designed to detect the willingness/preference and severity of third-party decisions. The 2 subcomponents were embedded in the decision and transfer phases, respectively. During the decision phase, participants decided whether to intervene in injustice events related to others. In the subsequent transfer phase, participants decided how many tokens to use to punish or help others if they chose to deduct the proposer or add the receiver, respectively. We constructed a computational model based on the assumption that participants made decisions by integrating self-payoff and other-regarding inequality aversion. This model allowed us to estimate the parameter values reflecting individual degrees of punishment and help, with a greater punishment severity or increased help requiring more tokens to diminish the violator's advantageous status or promote the victim's disadvantageous status, respectively. We expected to observe neural changes in brain functioning related to the theory of mind (e.g., ACC and TPJ) and subjective value computations (e.g., vmPFC and PCC) [31] under stress. This is because determining the severity of punishment or the extent of assistance is closely tied to considerations of others (such as the payoff and emotions involved) and the evaluation of available options in terms of their value.

## Results

### Acute stress induction with psychological, physiological, and endocrinal measures

We first examined the effectiveness of CPT-induced acute stress by analyzing salivary cortisol, heart rate, and self-reported affect feelings in the stress and control groups. We conducted two-way analyses of variance (ANOVA) of Group (between-subject factor: stress versus control) × Time points (within-subject factor) on salivary cortisol level, negative affect, and heart rate. This analysis revealed a significant Group × Time interaction for salivary cortisol level ($F(2.22, 110.75) = 11.231$, $P < 0.001$, $\eta_p^2 = 0.183$), self-reported negative affect ($F(2, 98) = 7.38$, $P < 0.001$, $\eta_p^2 = 0.131$), and heart rate ($F(1, 49) = 18.19$, $P < 0.001$, $\eta_p^2 = 0.271$). Bonferroni-corrected post hoc $t$ tests revealed that during CPT, the stress group exhibited higher levels of salivary cortisol ($P < 0.001$, **Fig 1B**) and negative feelings ($P < 0.001$, **Fig 1C**) than the controls and also exhibited elevated heart rates ($P < 0.01$, **Fig 1D**). These results indicate that CPT induced a prominent acute stress response, aligning with previous studies' findings using a similar stress paradigm. We compared relevant individual personality (e.g., perceived stress, empathetic concern) and endocrine variables (i.e., baseline basal testosterone and oxytocin level) between groups to exclude other possible individual effects that might confound the acute stress effect. These comparisons revealed no significant differences in these measures (all $P > 0.131$;

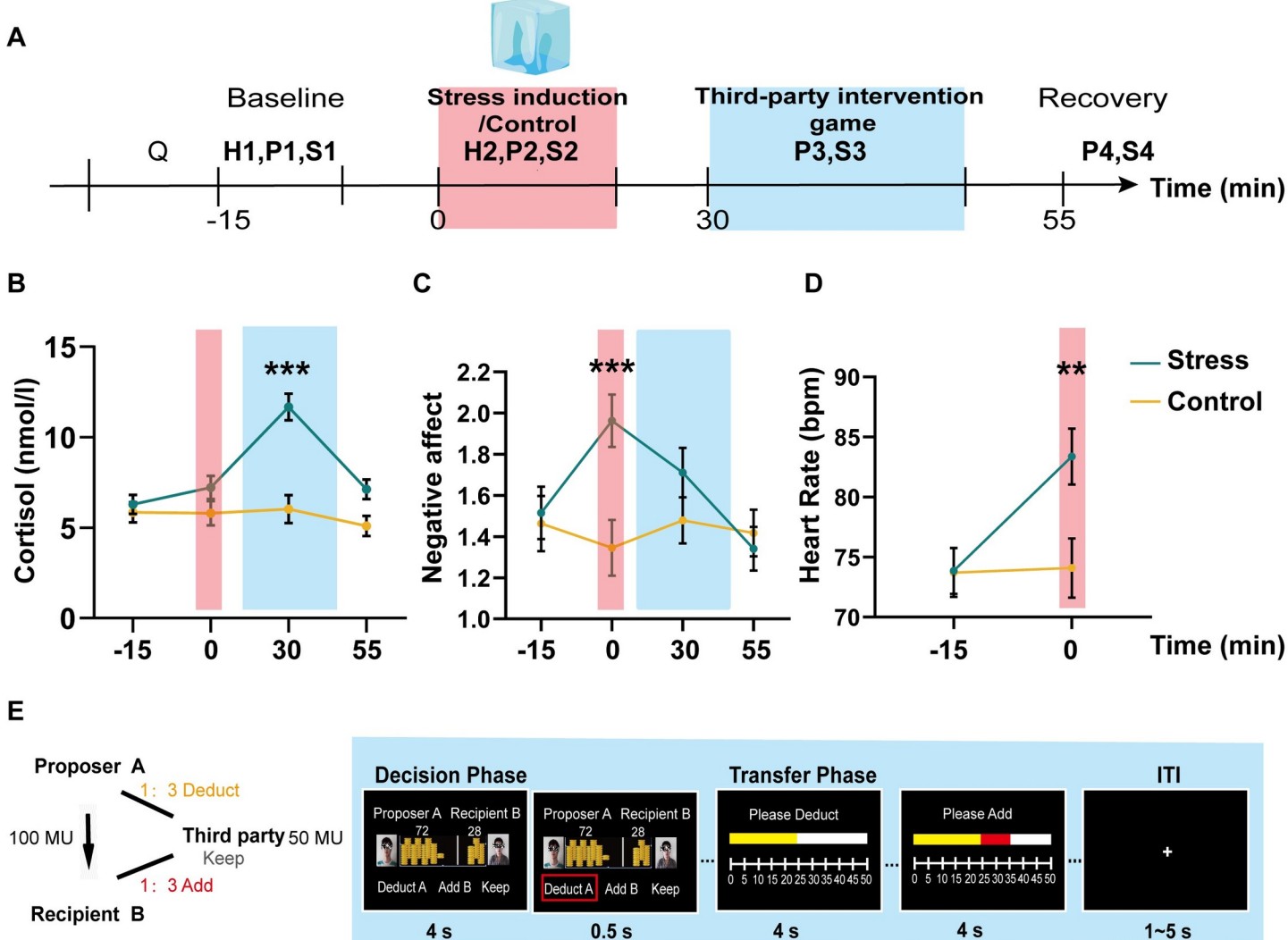

**Fig 1. Experimental procedure, stress induction, and TPIG. (A)** Timeline depicting the experimental procedure. Four saliva samples with concurrent subjective affect ratings were collected throughout the task (S1–S4 and P1–P4). Heart rate was collected during stress induction (H1) and the TPIG (H2). **(B–D)** The curves depict salivary cortisol concentration, negative affect, and heart rate in the control and stress groups. The red and blue boxes represent the CPT and TPI time windows. **(E)** An illustration of the TPIG with a sample trial. In each trial, participants were asked to choose from 3 options within 4 s (i.e., subtract "A," add "B," or keep) in the decision window, and they were instructed to decide how many MUs to use to reduce the MUs of A or increase the MUs of B within 4 s in the transfer window. If the participants chose to keep the money for themselves, the transfer phase windows were shown to the participants, but they were asked to select the zero option (meaning no tokens were used to intervene in other-regarding events). A 1–5 s intertrial interval was set between each window to dissociate the neural signal from each phase. ***$P < 0.001$; **$P < 0.01$; error bars represent SEM. The source data of Fig 1B–1D can be found at https://osf.io/fkae9/. CPT, cold pressor test; MU, monetary unit; TPI, third-party intervention; TPIG, third-party intervention game.

see **S1 Table**). Thus, any observed differences in behavioral and brain responses between groups could be attributed to acute stress but not personality and baseline endocrine levels.

### Acute stress affects third-party decisions between punishment and help in response to inequality events

We next investigated the effect of acute stress on behavioral performance in the TPIG. We conducted a two-way ANOVA of Group (between-subject factor: stress versus control) ×

Fairness (within-subject factor: 50:50 versus 60:40 versus 70:30 versus 80:20 versus 90:10) × Intervention type (within-subject factor: punish versus help) regarding the likelihood of each choice in the decision phase. The analysis conducted revealed a lack of statistical significance in the Group × Fairness × Intervention interaction ($F (3, 150) = 2.811$, $P = 0.108$, $\eta_p^2 = 0.040$). A further exploratory analysis revealed that acute stress affects decision-making only in extremely unfair conditions (i.e., 80:20 and 90:10 trials), with an increase in third-party decisions to help ($P < 0.05$ in 80:20 and 90:10 conditions) but a decrease in third-party decisions to punish ($P < 0.05$ in 80:20 and 90:10 conditions) in the stress group compared with the control group (**Fig 2A**). The parallel analysis of third-party intervention severity data revealed a similar pattern only in the extremely unfair condition. That is, acute stress decreased the third party's willingness to punish the norm violator ($P < 0.01$ in the 80:20 condition and $P < 0.05$ in the 90:10 condition) but had no effect on helping the receiver ($P > 0.05$) (**Fig 2B**). We restricted our further analyses to focus on the stress-by-intervention type in highly unfair conditions.

We conducted two-way ANOVA of Group × Intervention type on the choice rate of the decision phase of extremely unfair trials and found a significant interaction ($F (1, 50) = 4.033$, $P = 0.05$, $\eta_p^2 = 0.075$). Bonferroni-corrected post hoc $t$ tests revealed that stress increased the likelihood of choosing helping ($t_{help} (50) = 2.11$, $P < 0.05$, 95% CI: 0.01, 0.31) but slightly decreased the likelihood of choosing punishment compared with the control group ($t_{punishment} (50) = -1.94$, $P = 0.058$, 95% CI: −0.30, 0.01) (**Fig 2C**). Using the same two-way ANOVA on the contribution in the transfer phase ($F (1, 50) = 3.13$, $P = 0.08$, $\eta_p^2 = 0.059$), we found that stress decreased the contribution made to punish the norm violator ($t_{help} (50) = 0.79$, $P = 0.44$, 95% CI: −2.06 to 4.71; $t_{punishment} (50) = -2.08$, $P < 0.05$, 95% CI: −6.01, −0.10) but had no effect on the contribution made to help the receiver (**Fig 2D**). These results indicate that acute stress when facing other extreme injustice events decreases third parties' willingness to punish norm violators and the severity of the punishment but increases their willingness to help victims. In addition, we conducted multivariate regression analyses to investigate the primary factor influencing stress-induced behavioral variation. Our findings revealed that cortisol played a significant role in shaping individual's behaviors. Specifically, we observed a positive correlation between cortisol levels and the proportion of helpfulness ($\beta = 0.09$, $p = 0.027$). Conversely, there was a negative correlation between cortisol levels and the proportion of punishment ($\beta = -0.09$, $p = 0.017$) (**S7** and **S8 Tables**).

## Acute stress alters the latent computations of third-party intervention behaviors

To further investigate the latent cognitive computations underlying how acute stress modulates third-party intervention behaviors, we constructed 4 plausible computational models (see Methods) to fit participants' behaviors in the transfer phase to model the mental computations of punishment severity and the extent of help. As shown in **Fig 2E**, the model comparisons revealed that our behavioral data could be best fitted by the other-regarding inequality aversion model. This model allows us to separate the latent computations regarding the extent of help (i.e., how averse a person is to observe someone else being hurt) and the severity of punishment (i.e., how averse a person is to observe someone else hurt others), which are quantified by parameters $\beta$ and $\alpha$, respectively. We computed the value of parameter α minus parameter β as an index of punishment bias, ranging from −1 to 1. A larger value indicates that individuals tend to punish the norm violator more severely than the extent to which they help the victim. The one sample $t$ test revealed that acute stress reduced the punishment bias compared to that demonstrated by the controls (**Fig 2F**, $t (50) = -2.62$, $P < 0.05$, 95% CI: −0.50, −0.07). We

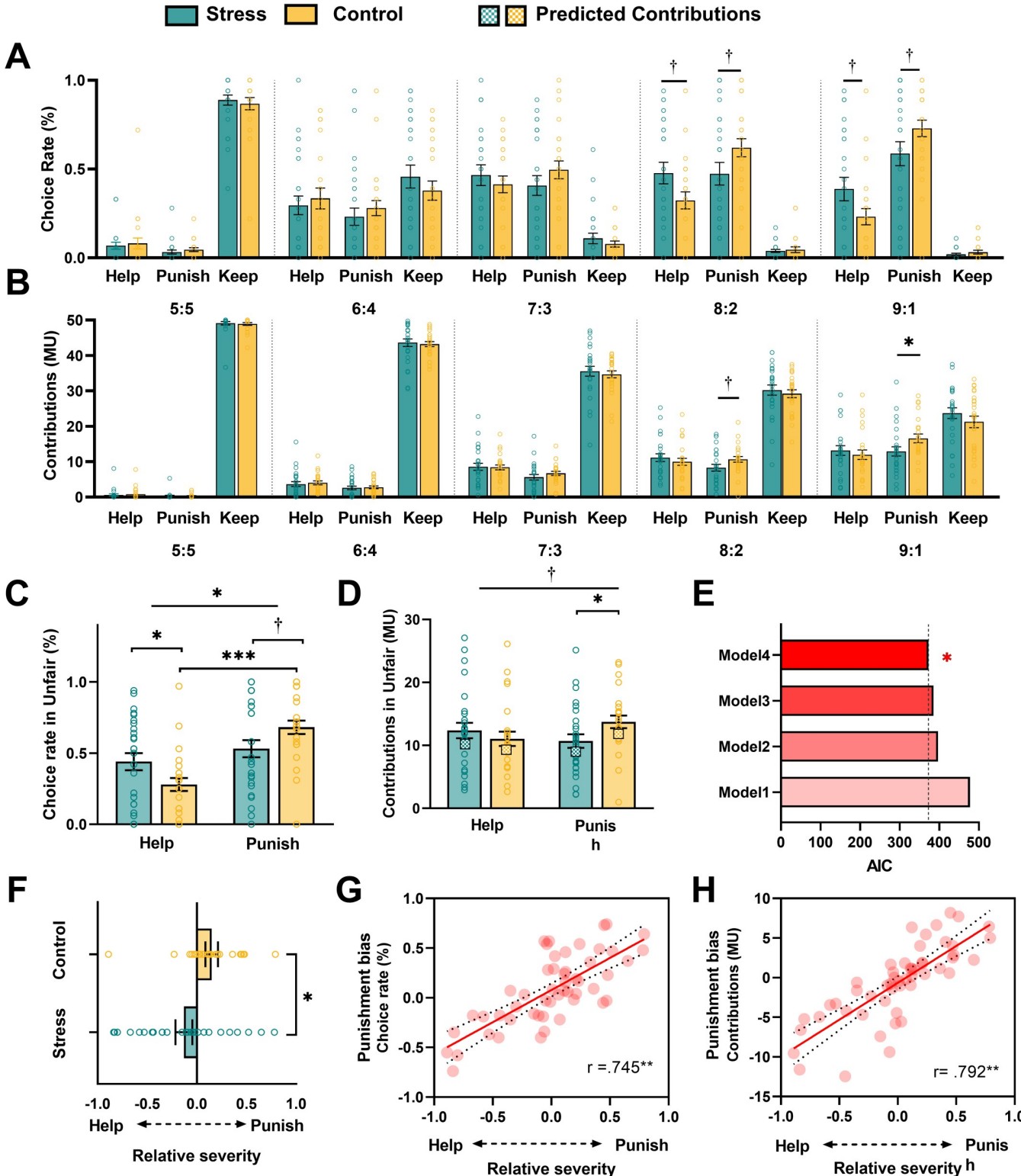

**Fig 2. Behavioral measures and computational modeling of the TPIG. (A and C)** Stress decreased the punishment rate but increased the help rate in the unfair condition (80:20 and 90:10). (**B**) In the unfair condition, stress decreased the severity of punishment. (**D**) Stress decreased the severity of punishment. The squares

represent the predicted contributions of punishment and help under unfair conditions based on winning model. (**E**) An overview of model comparisons: Models 1 and 2 represent the baseline model and the inequality aversion model. Model 3 represents the other-regarding inequality aversion model with a shared parameter of relative severity preference. Model 4 represents the other-regarding inequality aversion model with 2 parameters of the severity of punishment and help separately. (**F**) The difference between the stress and control groups in the relative severity preference based on the computational model ($\alpha-\beta$). (**G**) The actual choice rate differences between punishment and help (punishment bias) in all conditions as a function of the model-estimated relative severity. (**H**) The actual contribution differences between punishment and help (punishment bias) in all conditions as a function of the model-estimated relative severity. Each dot represents the data of a single participant. ***$P < 0.001$; **$P < 0.01$; *$P < 0.05$; †$P < 0.01$. Error bars represent the SEM. The source data of Fig 2A–2H can be found at https://osf.io/fkae9/. MU, monetary unit; TPIG, third-party intervention game.

further examined the association between model-based parameters and model-free behavior outputs to confirm the psychological meaning of the punishment bias value. We found a positive correlation of the punishment bias value with the relative punishment rate in the decision phase (i.e., the frequency of choosing punishment minus that of choosing helping, **Fig 2G**, r = 0.745, $P < 0.01$) as well as with the relative punishment severity in the transfer phase (i.e., the contribution donated to punishment minus the contribution donated to helping, see **Fig 2H**, r = 0.792, $P < 0.01$). These results indicate that acute stress reduces punishment bias by decreasing punishment severity but increasing the extent of help.

## Stress increases amygdala and prefrontal network integration in third-party intervention responses to inequality events in the decision phase

To investigate the overall effect of acute stress on brain systems involved in the TPIG, we conducted a general linear model (GLM) with a parametric design [32] to identify brain regions coding "Inequity" on the decision phase and the overall effect induced by stress. We initially conducted a whole-brain correction and observed increased activation in the right TPJ and right cerebellum in the stress group compared to the control group (**S2 Table**, voxel level threshold $P_{\text{uncorrected}} < 0.001$; cluster level threshold $P_{\text{FWE}} < 0.05$, whole-brain corrected). To further focus on specific brain regions related to acute stress, we applied a small-volume correction (SVC). This analysis revealed that the stress group exhibited higher activation in the left amygdala and bilateral insula compared to the control group (initial threshold $P < 0.001$; extent threshold of $P_{\text{FWE}} < 0.05$). These results confirm earlier findings indicating the essential role of the emotional salience network in the acute stress response, indicating the validity of CPT-induced changes in brain functioning [16]. Critically, we found that the activity in the rTPJ, VLPFC, and DLPFC was positively correlated with distributional inequity between the proposer and the recipient for both the stress and control groups (S3 Table, voxel level threshold $P_{\text{uncorrected}} < 0.001$; cluster level threshold $P_{\text{FWE}} < 0.05$, whole-brain corrected). These findings support the meta-analytical result on the neural processing of inequity by Feng and colleagues [33]. To further investigate other regions of interest (ROIs) related to inequity, we performed additional SVC. Our results revealed a significantly higher correlation between the degree of distributional inequity and the activity in the right amygdala among control individuals compared to stressed individuals (**Fig 3A** and **S3 Table**).

To further investigate the effects of acute stress on the amygdala-centric emotional salience network, we conducted a whole-brain psychophysiology interaction (PPI) analysis using the amygdala as a seed of interest. This analysis revealed higher functional connectivity of the right amygdala with the vmPFC in response to trials where participants chose the punishment option in the stress group but not in the control group (**Fig 3A and 3B** and **S5 Table**, voxel level threshold $P_{\text{uncorrected}} < 0.001$; cluster level threshold $P_{\text{FWE}} < 0.05$, whole-brain corrected). Using a mediation model, we further found that acute stress reduced the punishment rate by increasing amygdala-vmPFC communication (**Fig 3C**). This result indicates that although stress decreases the insensitivity of the right amygdala to the degree of inequity, it

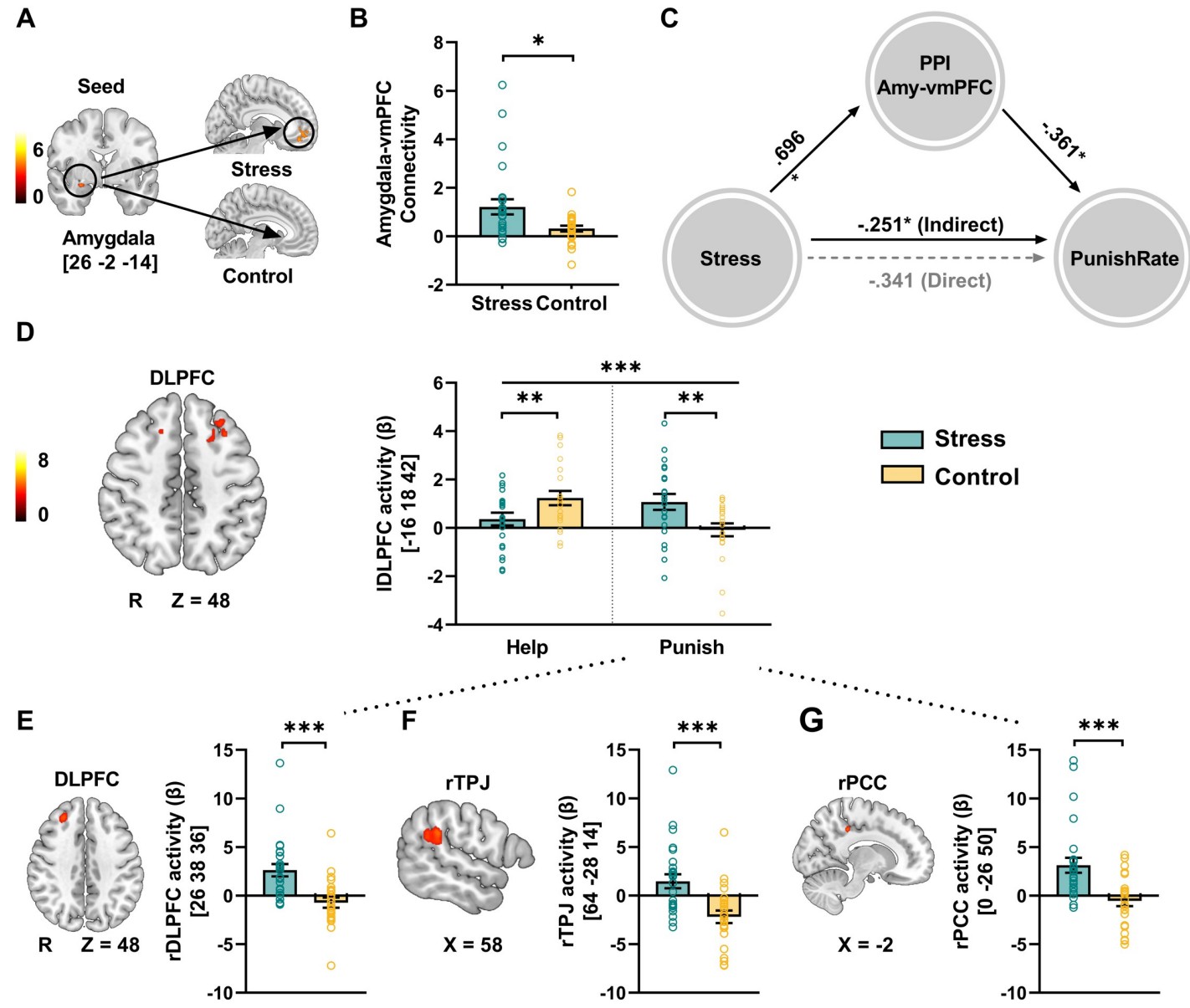

**Fig 3. Stress-induced changes in amygdala emotional salience and prefrontal executive networks in the decision phase. (A and C)** Stress-induced differences in brain representation of the degree of inequality and the functional connectivity between the amygdala and vmPFC. **(A and B)** PPI analysis based on the right amygdala as a seed showed that the stress group had increased functional connectivity of the right amygdala and the vmPFC during the decision stage in the punishment options. *$P < 0.05$; error bars represent the SEM. **(C)** The mediating effect of amygdala-vmPFC connectivity on the association between acute stress and the punishment rate (i.e., the frequency of selecting the punishment option in the decision phase). **(D–G)** Stress-induced neural activity in the decision phase. **(D)** Relative to the control group, the stress group demonstrated stronger DLPFC activation when selecting the punishment option than when selecting the help option (initial whole-brain threshold $P < 0.001$, cluster corrected $P_{FWE} < 0.05$ for left DLPFC). **(E–G)** Relative to the control group, the stress group had stronger activation in the rDLPFC, rTPJ, and rPCC in trials in which participants selected the punishment option (initial threshold $P < 0.001$; cluster corrected $P_{FWE} < 0.05$), and the effect was not significant in trials in which participants selected the help option. The source data of Fig 3B–3G can be found at https://osf.io/fkae9/. DLPFC, dorsolateral prefrontal cortex; PPI, psychophysiology interaction; vmPFC, ventromedial prefrontal cortex.

fundamentally changes the communication between the amygdala and the emotion regulation network and further decreases the frequency of choosing the punishment option in the decision phase.

Moreover, we compared brain activation maps for trials in which punishment was selected with trials in which help was selected between the stress and control groups. In the control group, we found higher activity in the DLPFC when selecting the helping option than when the punishment option was selected (**Fig 3D**). However, the stress group showed an inverse pattern: higher activity in the DLPFC was observed when selecting the punishment option than when selecting the helping option (**Fig 3D**, voxel level threshold $P_{\text{uncorrected}} < 0.001$; cluster level threshold $P_{\text{FWE}} < 0.05$, whole-brain corrected). We also compared the punishment and help choices separately between the stress and control groups. This contrast revealed that only during the punishment choices did acute stress induce higher activity in the right DLPFC, right TPJ (rTPJ), and right PCC (rPCC) (**Fig 3E–3G**, voxel level threshold $P_{\text{uncorrected}} < 0.001$; cluster level threshold $P_{\text{FWE}} < 0.05$, whole-brain corrected). These results indicate that punishment activates the DLPFC, rTPJ, and rPCC to a greater extent in the stress group than in the control group. However, the parallel analysis of brain activity and connectivity associated with the help trials revealed no clear effects when comparing the acute stress and control groups.

## Acute stress alters brain functional connectivity and latent computations in the transfer phase

To investigate the neurocomputational mechanisms of how acute stress alters functional brain systems involved in third-party intervention responses to inequality events, we first conducted a parametric modulation analysis with trial-by-trial subjective values (the "Utility" in the computational model) as a parametric modulator during the transfer phase. This analysis identified that the vmPFC, PCC, and several other regions were critical for the integrated subjective value of the transfer magnitude by maximizing utility for both stressed and control groups (**Fig 4B** and **S6 Table**, voxel level threshold $P_{\text{uncorrected}} < 0.001$; cluster level threshold $P_{\text{FWE}} < 0.05$, whole-brain corrected). However, after applying correction for multiple comparisons, no significant differences were found in brain regions associated with subjective value representation between the stress and control groups.

Motivated by the behavioral effect on the stress-induced decrease in prosocial punishment in the transfer phase, we investigated the neurocomputational correlates of punishment severity bias (punishment relative to help, $\alpha$-$\beta$) during the transfer stage. Through whole-brain multiple regression analysis, we discovered that punishment severity bias exhibited a stronger correlation with the activity of the PCC and rTPJ in stressed versus control participants (**S6 Table**, voxel level threshold $P_{\text{uncorrected}} < 0.001$; cluster level threshold $P_{\text{FWE}} < 0.05$, whole brain corrected). Additionally, SVC revealed that punishment severity bias displayed a stronger correlation with the activity of the ACC in stressed participants compared to control participants (**S6 Table**, voxel level threshold $P_{\text{uncorrected}} < 0.001$; cluster level threshold $P_{\text{FWE}} < 0.05$, SVC corrected). However, the amygdala and insula, which are highly associated with stress, did not exhibit differences between the stress and control groups in this context. To further analyze the aforementioned notable brain regions, we discovered a potential reversal between stress and control group during the neurocomputational of punishment severity bias. Specifically, in the stress group, there was a positive association between punishment severity bias and the activity of rACC ($r = 0.383$, $p = 0.053$, marginally significant) and rPCC ($r = 0.633$, $p < 0.001$). Conversely, in the control group, there was a negative association between punishment severity bias and the activity of rACC ($r = -0.519$, $p = 0.008$) and rPCC ($r = -0.610$, $p = 0.001$) (**Fig 4A**).

We further conducted a moderated mediation model to confirm that acute stress moderated the pathway from neural activation to behavioral outputs. In this model, the activation of

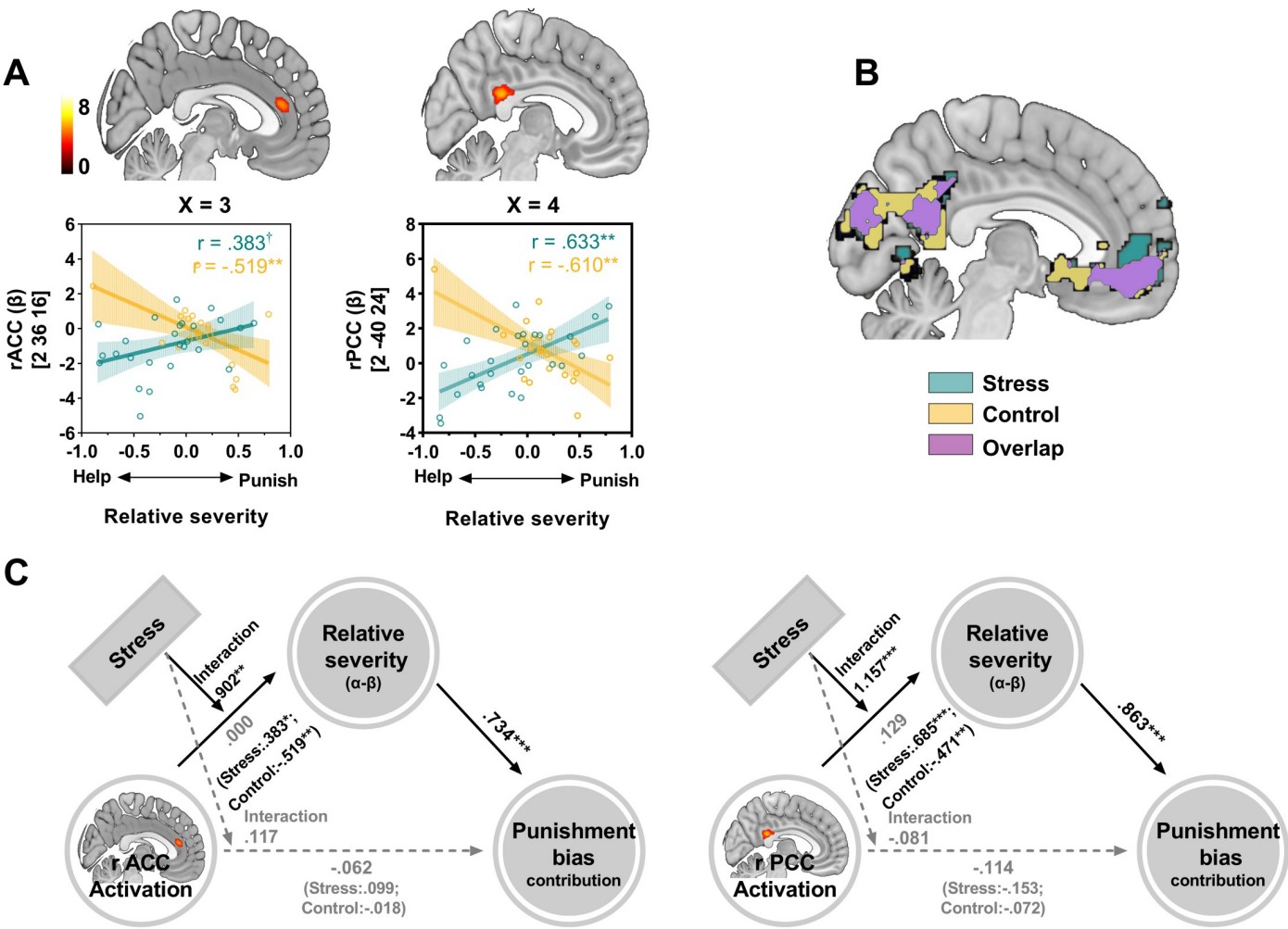

**Fig 4. Stress-induced changes in brain-behavior associations with severity computations in the transfer stage.** (**A**) Brain-behavior relationships between the stress-induced shift of punishment bias ($\alpha$-$\beta$) and activation in the rACC and PCC. Scatter plots depict the correlations between punishment bias ($\alpha$-$\beta$) and activation in these regions. (**B**) Parametric modulation with trialwise Utility revealed a significant correlation with BOLD responses in the vmPFC and PCC for both the stress and control groups. (**C**) The moderated mediation models depict that the punishment bias ($\alpha$-$\beta$) could account for the indirect associations of value computation in the activity of the rACC (left panel) and rPCC (right panel) with the punishment severity bias (punishment contribution minus help contribution). Acute stress moderated the correlation between value computation and punishment bias. All significant clusters were determined by a voxel level threshold $P_{\text{uncorrected}} < 0.001$; cluster level threshold $P_{\text{FWE}} < 0.05$, SVC. The source data of Fig 4A and 4C can be found at https://osf.io/fkae9/. BOLD, blood oxygen level-dependent; SVC, small-volume correction; vmPFC, ventromedial prefrontal cortex.

the ACC (and PCC) in the transfer stage was used as an independent variable, punishment severity bias ($\alpha$-$\beta$) was used as a mediator, stress manipulation (stress or control) was used as a moderator, and punishment contribution bias (punishment contribution minus help contribution) was used as a dependent variable. This analysis revealed that acute stress acts as a moderator variable in a mediation model showed that the activation of the ACC and PCC affects behavioral performance through its effect on punishment bias (**Fig 4C**). Specifically, the stressed participants recruit more ACC and PCC cognitive resources for a larger punishment bias and impose a higher punishment contribution. In contrast, individuals who did not experience acute stress imposed a higher punishment contribution, as they showed a larger

punishment bias with lower ACC and PCC resource consumption. These results indicate that acute stress moderates the mediatory role of latent computations on the association between neural activity and behavior outputs.

## Discussion

This study investigated the behavioral and neurocomputational substrates of how acute stress impacts a third party's intervention in response to injustice events. Behaviorally, acute stress led to a decrease in the third party's willingness to punish the violator and the severity of the punishment but an increase in their willingness to help the victim. Computational modeling further revealed a shift in intervention severity bias from punishment to help under acute stress. The neuroimaging results demonstrate that acute stress leads to higher activity in the insula, amygdala, and TPJ, reflecting a hyperactive state of the emotional salience network, as observed in previous studies [16]. Critically, we found that stress altered the preference for punishment and helping by redistributing brain activity associated with value calculation and mentalizing. These findings will be discussed in a framework of behavioral and neurocomputational mechanisms of how acute stress affects third parties' decisions acting upon emotional salience, prefrontal control, and mentalizing networks.

On a behavioral level, our findings support the "tend-and-befriend" theory, and a growing body of studies has demonstrated that stressed individuals have greater altruistic (more trust, sharing, and generosity) [23,24], cooperative, and other-oriented tendencies [34]. Evolutionally, an appropriate and modulated stress response is the core of survival, and a "tend-and-befriend" strategy can lead to a good reputation and promote self and offspring safety [7]. In contrast, punishment entails a cost for both the punisher and the punished, and it is expensive and inefficient. The threat of retaliation and vengeance from the target might lead individuals to avoid punishment when other nonconfrontational options are available, especially in uncontrollable stress situations [35,36]. From the perspective of reciprocity, helping an individual in need might create opportunities for direct reciprocity from that individual or an uninvolved bystander [37,38]. This aligns with the notion that acute stress may trigger a social approach, motivating individuals to build social bonding with other social members.

By leveraging computational modeling, we modeled the process by which participants integrated value utility and assigned different mental weights and subjective biases to the allocation of amounts to punishing the offender and helping the victim when making a norm recovery. The model parameter showed that stressed participants have smaller punishment bias. That is, when subjective value utility calculations are made in stressful situations, people devote more of their psychological preferences to deciding how to help the victim than how to punish the offender.

On a neuroimaging level, we found that stressed individuals demonstrated increased DLPFC activity when selecting punishment options instead of help options, which is commonly associated with top-down executive function in controlling selfishness-related impulses [28], social norm compliance in adjusting inequity aversion [39], and integrating distinct information streams for appropriate decisions [40]. Particularly, we found that stressed individuals demonstrate greater DLPFC, PCC, and rTPJ activation in those punishment options, and no consistent region is affected by stress in helping options. Research has shown that the PCC plays an important role in value-based decision-making, especially in terms of behavioral control and the evaluation of affordances [41,42]. Numerous studies have also linked neural activity in the rTPJ , PCC and DLPFC with mentalizing-related computations [43], strategic choice [44], and executive functioning vulnerable to stress [62,63]. Based on the biphasic-reciprocal model in which behavior patterns transition from deliberate to intuitive under

stress, these results suggest that punishment in stressful situations reflects a relatively more complex aspect, which not only requires one to activate the mentalization function to consider the emotional state of the person in the situation but also requires investing cognitive resources in value calculation to weigh the pros and cons. Our results have provided a possible explanation, to some extent, that third-party punishment may require more deliberation compared to helping others. It entails greater cognitive control and reliance on calculations. This thoughtful and pro-social choice is temporarily impeded under stressful conditions.

Broadly, people typically prefer to punish the offender than help the victim for justice restoration [4,5]. Under stress, the previously observed considerate response underwent a significant transformation, and the effectiveness of punishment as a relatively desirable strategy for addressing instances of injustice involving others was diminished.

In conjunction with local activation, we observed that the activity in the rTPJ, VLPFC, and DLPFC was positively correlated with distributional inequity between the proposer and the recipient for both the stress and control groups, this corresponds to the early findings about inequity process [33]. Additionally, we discovered a stronger correlation between right amygdala activity and the degree of distributional inequity in the control group, in contrast to the stress condition. However, it is important to note that previous research has demonstrated the predictability of degree of inequity aversion from amygdala activity [45–47]. Our results indicate that stress diminishes the processing of negative emotions associated with unfair situations. Besides, we found that stress increased amygdala-vmPFC functional connectivity under the punishment options, previous research on third-party punishment has also discovered a strengthening of amygdala connectivity with lateral prefrontal regions that are involved in punishment decision-making [48]. Amygdala-vmPFC connectivity has previously been found to play a crucial role in reward- and value-based decision-making [49,50], as well as in the generation and regulation of negative emotions [51], and various aspects of social cognition, such as facial emotion recognition, theory-of-mind ability, and processing self-relevant information [52]. Our findings suggest that acute stress may lead to a reduction in general activation in the amygdala's response to inequity. However, when it comes to complex third-party punishment decision, stress may induce hypercoupling of brain networks involved in value computations, regulation of aversive emotions, and mentalizing.

Furthermore, we found that the vmPFC, PCC, and several other regions represented the subjective utility value in the transfer stage in both the stress and control groups. This finding is consistent with the results of a large body of studies in that acute stress did not fundamentally change the subjective value computation circuits [50,53]. We also found that stressed individuals demonstrated stronger rACC, rPCC, and rTPJ activities when computing the relative severity of punishment against help than those in the control group. The ACC has been shown to balance the motivational conflict between the immediate emotional reaction and maximize profit in the decision-making context [28,54]. Moreover, we found that acute stress plays a moderating role in the direction of value computation of the rACC and rPCC to the relative preference between punishment and help and subsequently affects the transfer level of punishment relative to help. Specifically, the stressed participants needed to use more cognitive rACC and rPCC resources for a larger punishment severity bias and to impose a greater punishment contribution. In contrast, individuals in the control group only need to invest a low level of rACC and rPCC resource consumption in deciding how severely to punish. These findings suggest that when individuals are under stress, they tend to exhibit a more deliberative response pattern towards punishment rather than helping. It appears that people allocate greater cognitive resources towards value-based computation when making decisions related to punishment. It is important to note that the sample collected for this study consisted only of males, so the results may be more applicable to men than women. Furthermore, given that a

physical stressor was employed in this study, it would be intriguing to investigate the impact of a social stressor, such as the Social Stress Test (TSST), on decision preferences. We are inclined to believe that a social stressor may engage the empathy and mentalizing networks to a greater extent when making decisions related to helping. Previous research has indeed shown that acute stress can enhance prosocial behavior by intensifying the sharing of others' emotions [18].

In conclusion, our study demonstrates a stress-induced shift in third parties' tendency from punishment toward help by acting on emotional salience, central-executive and theory-of-mind networks, characterized by higher amygdala activity and greater connectivity with the vmPFC, an increase in dorsolateral prefrontal engagement, and connectivity of the mentalizing network in relation to the stress-induced severity bias of punishment. Our findings suggest a neurocomputational mechanism of how acute stress reshapes third parties' decisions by reallocating value computations and neural resources in emotional, executive and mentalizing networks to inhibit punishment bias.

## Materials and methods

### Participants

Fifty-three male volunteers were recruited and randomly assigned to either the stress condition (CPT) or the control condition; 1 participant was removed from the analyses, as he did not understand the task. The final sample included 52 participants (stress: $n = 27$). Participants who met the following conditions were excluded: hormonal contraception, prescription or drug consumption, smoking, alcohol abuse, a history of chronic disease or mental condition, and a major evaluation within 2 weeks. Furthermore, participants were instructed to abstain from physical activity, meals, and caffeine consumption for 2 h prior to the study. The protocol was designed and performed according to the principles of the Helsinki Declaration and approved the Institutional Review Board of Department of Psychology at Renmin University of China (IRB2017052701). All participants signed written informed consent.

### Experimental procedures

Testing sessions took place in the afternoon (1:30–5:30) to control variability in diurnal cortisol secretion [62]. Upon arrival at the lab, participants completed questionnaires (psychological tests, see SI Appendix) for 20 min. Subsequently, the baseline heart rate (HR1) was recorded for 3 min, and the saliva sample (S1) and the Positive and Negative Affect Scale (PANAS, PA1, and NA1) were measured. Then, the participants were randomly assigned to CPT or control conditions. Heart rate (HR2) was recorded across the whole CPT for 3 min. The saliva sample (S2) and the PANAS (PA2 and NA2) were collected immediately after the CPT or a control manipulation. Then, after 10 min of rest and 10 min of T1 structural image collection in the MRI scanner, the subjects completed the first run of the third-party intervention task (TPI). The third saliva sample (S3) and the PANAS (PA3 and NA3) were measured 25 to 30 min after the CPT (run 1 for 10 min). The participants then completed the remaining 2 runs of TPI in the MRI scanner for 20 min. The fourth saliva sample (S4) and the PANAS (PA4 and NA4) were measured 50 min after the CPT (**Fig 1A**).

### Stress manipulation

Stress induction involved a CPT wherein participants submerged their left hand to the wrist in 0 to 4˚C ice water for 3 consecutive minutes [55]. The participants in the control group submerged their left hand in warm water (35 to 37˚C) for 3 consecutive minutes.

## Third-party intervention task

This task included 3 players. Player A (proposer) received an endowment of 100 monetary units (MUs) per round and could decide how to distribute it between himself/herself and player B (recipient) in units of 5 MUs (i.e., 0, 5, 10, 15, and 20). Player B was required to passively accept the proposal. Player C (the third party) received an endowment of 50 MUs (10 MUs = 1 Chinese yuan) per round and was instructed to observe the collection between player A and player B. In the fMRI scanner, participants believed that they were randomly assigned to player C via a massive drawing process. In the decision stage, participants were presented with 3 options: transferring MUs to decrease player A's MUs, transferring MUs to increase player B's MUs, or keeping all the MUs for themselves. If participants chose to reduce player A's money during the decision stage, they could first decide on the amount of MUs to transfer from their own initial 50 MUs and deduct that amount from player A's MUs in increments of 5 MUs in the first transfer stage. Following that, participants could further decide on the amount of MUs to transfer from their remaining MUs to compensate player B in the second transfer stage. On the contrary, if they chose "increase player B" during the decision stage, they could first determine the amount of MUs to transfer to player B during the first transfer stage and then proceed to choose a reduction in player A's MUs during the second transfer stage (see Fig 1E). It is important to note that the MUs transferred by participants will be applied to player A or player B at a triple ratio. For example, if they transfer 5 MUs to increase player B, player B will actually receive an increase of 15 MUs. Each run of the experiment consisted of 30 preprogrammed trials, and a total of 3 runs were conducted. The offers presented to the participants were specifically crafted, with average offer ratios ranging from 90/10 to 50/50. However, the actual offers shown during the experiment varied between 1% and 2%, for example, 91/9 and 88/12. The composition of fairness levels in the offers was not disclosed to the participants. Participants were aware that their decisions would not cause player A's income to become negative, as they were informed that their choices only affected the "floating" portion of the income for players A and B, not the "base" portion.

## Physiological measures

Saliva cortisol samples were obtained from participants at 4 time points before and after the stress/control manipulation (Sarstedt, Rom-melsdorf, Germany). These samples were stored at −80˚C until the analysis. We also measured testosterone and oxytocin levels in the saliva to rule out other hormonal effects [56] (see SI in appendix). The samples were thawed and centrifuged for 5 min at 3,500 rpm. An electrochemiluminescence immunoassay was used to measure salivary cortisol concentrations (Cobas e 601, Roche Diagnostics, Numbrecht, Germany). The intra- and interassay variations for cortisol were below 10%.

Furthermore, as a marker of sympathetic nervous system activity, heart rate was continuously monitored in the CPT stage (3 min) with a Polar WearLink + heart rate monitor (POLAR RCX3) to determine the effects of the stress induction versus the control task; heart rate was also monitored for 3 minutes as a baseline before the CPT (Fig 1A).

## Data analysis

**Behavioral data analysis.**   To assess the efficacy of the stress manipulation, mixed two-way ANOVAs were performed on salivary cortisol, heart rate, and subjective affective state, with stress manipulation (stress versus control) as the between-subjects variable and "time" point of measurement as the within-subject factor.

In the third-party intervention task, the rates and costs that subjects chose to punish A, help B or serve themselves were determined in 5 fair conditions. Since selfish choices were made

only in the 50:50 fair condition (**S1 Fig**), we focused on the trade-off between help and punishment choices in the subsequent analysis. The number of MUs and rates of punishing A or helping B were subjected to mixed three-way repeated measures ANOVA with stress manipulation as a between-subjects factor and option (help, punish) and fair condition (50:50, 60:40, 70:30, 80:20, 90:10) as 2 within-subject factors. Furthermore, we combined 6:4 and 7:3 as relatively unfair conditions (RUF), 80:20 and 90:10 as extreme unfair conditions (EUF), and 50:50 as the fair condition (F). The clustering criterion was established based on the findings of a previous meta-analysis [57]. Repeated measurement ANOVA was used in 2 conditions (RUF and EUF).

The Greenhouse–Geisser correction was applied when the requirement of sphericity in the ANOVA for repeated measures was violated. Effect sizes are expressed as partial eta-squared ($\eta_p^2$). Behavioral data were analyzed using MATLAB R2019a (Mathworks) and SPSS 25 (IBM).

## Computational modeling procedures

To better understand the neural mechanisms underlying acute stress and its impact on third-party intervention behaviors, we employed computational modeling to analyze the behavioral data. This approach enabled us to evaluate the extent to which unrelated third parties were personally invested in addressing inequality aversion. We separated the subcomponents of the extent of help (i.e., how averse an individual is to observe someone else being hurt) and punishment severity (i.e., how averse an individual is to observing someone else hurt others), denoted by parameters $\beta$ and $\alpha$, respectively. The values of the 2 parameters ranged from 0 to 1, with values close to 1 indicating greater magnitude of punishment or help. This model was referred to as the "**Other-regarding inequality aversion model**" since it hypothesizes that participants' actions were influenced by aversion to the inequity between the violator and the recipient. The model was formalized as follows:

$$
\begin{aligned}
U_{i(t)}&\left(S_{i(t)}^p, S_{i(t)}^h \mid \alpha_i, \beta_i\right) \\
&= \pi_{i(t)} - \left\{ \max\left[\alpha_i \cdot \left(x_{1(t)} - 50\right) - 3S_{i(t)}^p, 0\right] - \min\left[\beta_i \cdot \left(x_{2(t)} - 50\right) + 3S_{i(t)}^h, 0\right] \right\}
\end{aligned} \quad (1)
$$

$$
\hat{S}_{i(t)}^p, \hat{S}_{i(t)}^h = \arg\max U_{i(t)}\left(S_{i(t)}^p, S_{i(t)}^h\right) \quad (2)
$$

where $i$ indicates individual; $U_i$ is Participant $i$'s decision utility of spending tokens on punishment and help at trial $t$; $S^p$ represents the amount of tokens participant used to punish the violator ($x_1$) and $S^h$ represents the amount of tokens participant used to help the victim ($x_2$); $\pi$ represents participant's payoff, in each trial $\pi = 50 - S^p - S^h$. In Eq 2, $\hat{S}_{i(t)}^p$ and $\hat{S}_{i(t)}^h$ is the number of tokens that optimizes the participant's subjective utility. Note that $\hat{S}_{i(t)}^p$ and $\hat{S}_{i(t)}^h$ depends not only on the observed level of punishment and help but also on the participant-specific parameters $\alpha_i$ and $\beta_i$, which are estimated by minimizing the sum of squared errors (RSS):

$$
RSS = \sum i \sum t\left[\left(S_{i(t)}^p - \hat{S}_{i(t)}^p\right)^2 + \left(S_{i(t)}^h - \hat{S}_{i(t)}^h\right)^2\right] \quad (3)
$$

## Model comparisons

To assess whether the "other-regarding inequality aversion model" provided a compelling explanation of participants' behavior, here, 3 alternative models were compared by calculating the Akaike information criterion (AIC) for each model [5].

**Model 1: Baseline model.** The baseline model proposes a rational assumption that participants made decisions only by maximizing their self-payoff. That is, the absence of inequality

aversion hypotheses and participant choice not to help and punish instead would be an optimized option. The utility function was as follows:

$$U_{i(t)} = 50 - S^p_{i(t)} - S^h_{i(t)} \qquad (4)$$

**Model 2: Self-regarding inequality aversion model.** This model assumed that participants made decisions by weighting the payoff inequality between themselves and others versus the violator (who always has more than or equal to 50 tokens), with participants confronting disadvantageous inequality aversion while comparing with receiver participants facing advantageous inequality aversion. The participants' goal was to eliminate inequality aversion by both punishment and help. The utility function was formalized as follows:

$$U_{i(t)}(S^p_{i(t)}, S^h_{i(t)}|envy_i, guilt_i) = \pi_{i(t)} - envy_i \cdot DI_t - guilt_i \cdot AI_t \qquad (5)$$

where DI is disadvantageous inequality and AI is advantageous inequality aversion, which is calculated as follows:

$$DI_t = \max[x_{1(t)} - 3S^p_{i(t)} - \pi_{i(t)}, 0] + max[x_{2(t)} + 3S^h_{i(t)} - \pi_{i(t)}, 0] \qquad (6)$$

$$AI_t = \max[\pi_{i(t)} - (x_{1(t)} - 3S^p_{i(t)}), 0] + max[\pi_{i(t)} - (x_{2(t)} + 3S^h_{i(t)}), 0] \qquad (7)$$

Note that in each trial $\pi = 50 - S_p - S_h$.

**Model 3: Other-regarding inequality aversion model with shared parameters for estimating the severity of punishment and help.** This model is similar to the other-regarding inequality aversion model introduced above, except that a single parameter denoted the magnitude of punishment and help, $\alpha$, which participants hold the same level of "how averse an individual is to observe someone else being hurt" and "how averse an individual is to observe someone else hurt others." In other words, participants used the same but distinct psychological mechanisms when determining whether to help or punish others.

**Model validation.** *Parameter recovery*. We ran parameter recovery analyses to ensure that our model was robustly identifiable. This approach allowed us to assess whether our model overfitted the data and to estimate the reliability of the model parameters. To this end, we created 52 simulated participants by simulating task data at random points within the parameter space and fit our model to these simulated subjects' behaviors.

*Model prediction*. We calculated the correlation between behavioral output predicted by our model and true behaviors. To this end, we first used the estimated parameters of 52 participants to generate predicted behaviors individually and then calculated the correlation between the number of tokens from actual observation in the task and that from predicted behaviors in both punish and help conditions.

## fMRI data analysis

**Imaging data acquisition and preprocessing.** Brain imaging data were acquired on a 3T Prisma MR scanner (Siemens, Germany). During the tasks, blood oxygen level-dependent (BOLD) signals were acquired with a prototype simultaneous multislice echo-planar imaging (EPI) sequence (echo time, 30 ms; repetition time, 2,000 ms; field of view, 224 mm × 224 mm; matrix, 112 × 112; inplane resolution, 2 mm × 2 mm; flip angle, 90 degrees; slice thickness, 2.0 mm; 62 slices; slice orientation, transversal). Field map images were acquired using a vendor-provided Siemens gradient echo sequence (gre field mapping: echo time 1, 4.92 ms; echo time 2, 7.38 ms; repetition time, 620 ms; flip angle, 60 degrees; bandwidth, 565 Hz/pixel) with the

same geometry and orientation as the EPI image. A high-resolution 3D T1 structural image (3D magnetization-prepared rapid acquisition gradient echo; 0.5 mm × 0.5 mm × 1 mm resolution) was also acquired. Image preprocessing was conducted using the Statistical Parametric Mapping package (SPM12, RRID: SCR_007037; Welcome Department of Imaging Neuroscience, London, United Kingdom). EPI volumes were realigned to the first volume, corrected for geometric distortions using the field map, coregistered to the T1 image, normalized to a standard template (Montreal Neurological Institute, MNI), resampled to $2 \times 2 \times 2$ mm$^3$ voxel size, and spatially smoothed with an isotropic 8 mm full width at half-maximum Gaussian kernel.

**General linear model.** We regressed the fMRI time series into 3 GLMs to investigate how acute stress affected the brain's decision circuitry. We looked for neural activity associated with inequity in the decision stage, i.e., the degree of inequity between the proposer and recipient (GLM_ Inequity) with the first GLM. We also checked the stress manipulation effect. In the second GLM (GLM_ Choice), we aimed to recognize brain regions with behavior associated with punishment and help choices under the unfair condition. The third GLM (GLM_ Utility) sought to identify brain regions coding utility and severity preference between punishment and help during the transfer stage. For the main contrasts, the individual voxel threshold was set to $P < 0.001$. We performed whole-brain corrections for multiple comparisons at the cluster level ($P_{FWE} < 0.05$).

Additionally, we conducted supplementary analyses to explore brain regions that have consistently emerged as relevant in previous studies and are aligned with our research design. These supplementary analyses involved SVC, which was applied after the whole-brain multiple comparisons correction. The purpose of this correction was to enhance the comparability of our results with those of prior studies. Specifically, when examining the effects of stress, we performed SVC on stress-related brain regions, including the amygdala and insula. Similarly, when investigating the processing of "inequity," we applied SVC to brain regions known to be involved in this process. The threshold for SVC was set at $P < 0.001$ for individual voxels, with a cluster level of significance at $P_{FWE} < 0.05$. We defined the anatomical ROIs for the bilateral amygdala, anterior cingulate cortex, and insula using the SPM Wake Forest University (WFU) Pickatlas toolbox (www.ansir.wfubmc.edu, version 3.0) [58]. The anatomical coordinates for the "inequity"-related areas were obtained from a previous meta-analysis. We used a sphere with a diameter of 8 mm as the size criterion for the ROIs, and the coordinates for these regions are provided in S9 Table.

In the first model (GLM_ Inequity), the "Inequity" δ was defined as "δ = |MU Proposer −MU $_{Recipient}$|" [59]. Four regressors were included in the GLM in the following order: (i) onset of the decision stage at the beginning of each trial when participants saw the distributions; (ii) parametric modulation of the trialwise "Inequity"; (iii) onset of the first transfer stage; and (iv) onset of the second transfer stage. We modeled BOLD responses at these onsets as stick functions. All regressors and 6 head movement regressors of no interest were convolved with a canonical hemodynamic response function. For each event, onset regressor parameter estimates were obtained, and contrast images of each of the parameters against zero were generated. The obtained images were transferred to a second-level random-effects analysis using a two-sample *t* test and conjunction analysis to compare the stress and control groups.

In the second model (GLM_ Choice), we focused on the decision window when participants responded to the trials of unfair distribution (8:2 and 9:1). We defined the following 2 onset regressors of interest: (i) onset of the punishment choice of unfair trials; and (ii) onset of the help choice of unfair trials. We also defined the following uninterested regressors: (iii) onset of the punishment transfer of unfair trials; (iv) onset of the help transfer of unfair trials;

(v) onset of all the choices of relatively unfair trials (7:3 and 6:4, for punishment, help and keep choice); (vi) all the transfers of relatively unfair trials (7:3 and 6:4, for punishment, help and keep choice); (vii) onset of all the choices of fair trials (5:5, for punishment, help and keep choice); and (viii) onset of all the transfers of fair trials (5:5, for punishment, help and keep choice). The GLM additionally included 6 movement regressors of no interest, 3 for translational movements (x, y, and z) and 3 for rotation movements (pitch, roll, and yaw). All regressors were convolved with the canonical hemodynamic response function. Individual contrast images (for "Punishment," "Help," "Punishment-Help") were transferred to a second-level random-effects analysis using two-sample $t$ tests to compare the stress and control groups.

In the third model (GLM_ Utility), we computed a GLM with a parametric design to identify brain regions coding "Utility" on the transfer window when participants spent chips (MU) to punish the proposer or help the recipient. Utility was defined in our "**Other-regarding inequality aversion model.**" Four regressors were included in the GLM in the following order: (i) onset of the first transfer stage; (ii) parametric modulation of the trialwise "Utility"; (iii) onset of the decision stage at the beginning of each trial; and (iv) onset of the second transfer stage. Similar to the first model, we modeled BOLD responses at these onsets as stick functions. All regressors and 6 head movement regressors of no interest were convolved with a canonical hemodynamic response function. The obtained images were transferred to a second-level random-effects analysis using two-sample $t$ tests and conjunction analysis to compare the stress and control groups.

**gPPI analysis.**    To investigate whether the functional connectivity of the amygdala differed during help and punishing decisions and whether it was affected by stress, we performed a whole-brain gPPI analysis with the right amygdala as the seed region [5]. The location of the right amygdala seed ROI was based on a 6 mm radius sphere centered at the peak activation within the contrasts of stress versus control (GLM_ Inequity). We estimated a GLM with the following regressors: (i) a physiological regressor (i.e., the entire time series of the seed region over the whole experiment); (ii) a psychological regressor for the onset of the punishment choices; (iii) the PPI regressor for the punishment choices; (iv) a psychological regressor for the onset of the help choices; and (v) a PPI regressor for the help choices. The onset and PPI regressors were convolved with a canonical form of the hemodynamic response. The model also included the 6 motion parameters as regressors of no interest. Individual contrast images for functional connectivity ("punishment," "help," "punishment vs. help") were transferred to a second-level random-effects analysis using a two-sample $t$ test and one-sample $t$ test.

**Mediation/moderation analysis.**    We used mediation analysis to test the hypothesis that stress affects punishment rates by amygdala-vmpfc functional coupling. We constructed a mediation model (Model 5) that specified stress manipulation (stress or control) as an independent variable, amygdala-vmpfc functional connectivity as a mediator, and punishment rate as a dependent variable. In addition, a moderated mediation analysis (Model 8) was conducted to test the hypothesis that punishment bias ($\alpha$-$\beta$) could account for the indirect associations of value computation in the activity of the rACC, rPCC, and rTPJ with punishment severity bias (punishment contribution minus help contribution), and acute stress moderated the correlation between value computation of the rACC, rPCC, and rTPJ and punishment bias. These analyses were performed by the SPSS "PROCESS," made available by Andrew Hayes [60]. A nonparametric resampling procedure (bootstrapping) with 5,000 samples was used to estimate the significance of the indirect effect. Through bootstrapping, we calculated point estimates of the indirect effects and constructed a 95% confidence interval around each point estimate.

## Supporting information

**S1 Text. Supplementary methods.**
(DOCX)

**S1 Fig. Acute stress modulated the prosocial preference.**
(DOCX)

**S2 Fig. Model validation and parameter recovery.**
(DOCX)

**S3 Fig. Stress influences the neural correlates of punishment versus help decision under all unfair conditions (relative and unfair condition, 60:40~90:10).**
(DOCX)

**S1 Table. Control variables per group [M (SD)].**
(DOCX)

**S2 Table. Stress-induced differences in the brain among all fair conditions in the decision stage.**
(DOCX)

**S3 Table. Neural response for increasing distributional inequity |Proposer-Recipient | in decision stage.**
(DOCX)

**S4 Table. Neural correlates of utility in transfer stage.**
(DOCX)

**S5 Table. PPI: Regions showing stress group functional connectivity with the right amygdala when making a punishment choice in the unfair condition.**
(DOCX)

**S6 Table. Stress induced higher neural correlates of relative severity in punishment versus help ($\alpha$-$\beta$).**
(DOCX)

**S7 Table. Linear relationship between stress-related variance and behavioral decision-making (Help Rate).**
(DOCX)

**S8 Table. Linear relationship between stress-related variance and behavioral decision-making (Pinish Rate).**
(DOCX)

**S9 Table. The neural correlates of total utility associated with aversion to witnessing someone else harm others ($\alpha$).**
(DOCX)

**S10 Table. The neural correlates of total utility associated with aversion to witnessing someone else being harmed ($\beta$).**
(DOCX)

## Author Contributions

**Conceptualization:** Huagen Wang, Shaozheng Qin, Chao Liu.

**Data curation:** Huagen Wang, Sihui Zhang, Zhenhua Xu.

**Formal analysis:** Huagen Wang.

**Funding acquisition:** Shaozheng Qin, Chao Liu.

**Investigation:** Huagen Wang.

**Methodology:** Huagen Wang, Xiaoyan Wu, Jiahua Xu, Ruida Zhu.

**Project administration:** Huagen Wang, Shaozheng Qin, Chao Liu.

**Supervision:** Shaozheng Qin, Chao Liu.

**Visualization:** Huagen Wang.

**Writing – original draft:** Huagen Wang.

**Writing – review & editing:** Huagen Wang, Ruida Zhu, Xiaoqin Mai, Shaozheng Qin, Chao Liu.

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
