## [Editor Report · Decision Letter 0]

7 Jun 2023

Dear Dr Qin, 

Thank you for submitting your manuscript entitled "Acute stress reshapes third-party punishment and help decisions: Behavioral evidence and neurocomputational mechanisms." for consideration as a Research Article by PLOS Biology.

Your manuscript has now been evaluated by the PLOS Biology editorial staff as well as by an academic editor with relevant expertise and I am writing to let you know that we would like to send your submission out for external peer review.

Once your full submission is complete, your paper will undergo a series of checks in preparation for peer review. After your manuscript has passed the checks it will be sent out for review. To provide the metadata for your submission, please Login to Editorial Manager (https://www.editorialmanager.com/pbiology) within two working days, i.e. by Jun 09 2023 11:59PM.

Kind regards,

Christian

Christian Schnell, PhD

Senior Editor

PLOS Biology

cschnell@plos.org

---

## [Decision Letter · Decision Letter 1]

6 Sep 2023

Dear Dr Qin,

Thank you for your patience while your manuscript "Acute stress reshapes third-party punishment and help decisions: Behavioral evidence and neurocomputational mechanisms" was peer-reviewed at PLOS Biology. It has now been evaluated by the PLOS Biology editors, an Academic Editor with relevant expertise, and by several independent reviewers. 

In light of the reviews, which you will find at the end of this email, we would like to invite you to revise the work to thoroughly address the reviewers' reports.

As you will see below, the reviewers agree that the study is overall interesting and potentially provides important insights. However, R1 raises concerns about empathy as a potential confounder and identifies other concerns about the computational modelling and the neuroimaging analysis. R2 has less serious concerns but mentions some previous studies that might impact novelty of this study. R3 raise concerns about the integration of the previous literature and some technical aspects about the computational and neuroimaging aspects of the study. Please bear the reviewer comments carefully in mind when revising the manuscript.

Please also note that we had asked the reviewers if they want to provide additional feedback after reading the other reviewers' reports. In this context, Reviewer 3 wrote:

"We don’t agree with reviewer 1’s comments on the computational modeling: 1) inequity is included in the full model (equation 1) because the payoffs of both players are included in the utility such that a higher inequity between the two players prompts the third-party to punish the transgressor more harshly and help the victim more, and 2) the authors used RSS to fit the model parameters but AIC to compare the models which takes care of the number of parameters."

We usually use these comments only to inform the discussion between the editorial team and the academic editors but in this case we thought this comment would be useful for you when revising your manuscript and asked the reviewer for permission to share these comments with you. 

Given the extent of revision needed, we cannot make a decision about publication until we have seen the revised manuscript and your response to the reviewers' comments. Your revised manuscript is likely to be sent for further evaluation by all or a subset of the reviewers.

**IMPORTANT - SUBMITTING YOUR REVISION**

*Re-submission Checklist*

*Published Peer Review*

*PLOS Data Policy*

*Blot and Gel Data Policy*

Sincerely,

Christian

Christian Schnell, PhD

Senior Editor

PLOS Biology

cschnell@plos.org

REVIEWS:

Reviewer #1: Wang et al. carried out an fMRI experiment with a newly designed third-party intervention game. They aimed to study the impact of the cold pressure test (CPT) on the intervention behavior and fMRI signals of the third party, utilizing a computational model. The authors discovered that CPT lowered the third party's readiness to penalize norm violators (proposers in unfair divisions) and the intensity of the penalty. However, it elevated their willingness to aid recipients. These behavioral patterns were further interpreted by their computational model.

In their fMRI analysis, Wang et al. first reported that the activity in the left amygdala (though the right amygdala exhibited an opposite pattern), bilateral insula, and rTPJ, correlated with the inequity between the proposer and recipient, was larger in the CPT group than in the control group. Nevertheless, rTPJ, VLPFC, and DLPFC were similar in both groups. Their subsequent PPI and mediation analyses revealed that in the CPT group, the functional connectivity of the right amygdala with the vmPFC was higher when choosing the punishment option, and CPT reduced the punishment rate by enhancing amygdala-vmPFC connectivity. Contrarily, DLPFC activity displayed a reverse pattern in the CPT and control groups, with higher DLPFC activity when choosing the punishment and help options, respectively. They also found that the correlation between punishment severity bias and the activity of ACC, PC, and rTPJ was stronger in CPT versus control participants.

The authors interpreted these results as showing that "acute stress decreases third-party's willingness to punish violators and the severity of the punishment, but increases their willingness to help victims by using an intuitive mode suggested by higher activity in the insula, amygdala, and TPJ, and smaller involvement of the amygdala-vmPFC network, DLPFC activity, and theory-of-mind network." If this interpretation is robustly validated, this study could provide a significant insight into social neuroscience.

However, the reviewer believes there are serious concerns in the areas of behaviors, computational modeling, and fMRI analysis. These need to be addressed to make the authors' interpretations persuasive. These are listed as follows:

1. Behaviors

The authors appear to presume that behavioral changes were a result of acute stress, particularly the impact of cortisol and the hypothalamic-pituitary-adrenal (HPA) axis. However, subjective and cognitive experiences of a negative event (even reading a story) can increase empathy. Some research has suggested that individuals who have faced adversities are more likely to exhibit helping behaviors (e.g., Decety, J. Ann N Y Acad Sci. 2011).

To counter this possibility, the authors need to separate the effects of cortisol increase, heart rate increase, and the stress questionnaire and show that the cortisol increase has the strongest influence on behavioral changes. Such analysis is possible because the correlation coefficient among these three stress variables is generally not very high (some show an clear increase in cortisol concentration, while others do not).

2. Computational modeling

(a) The authors used the inequity between proposer and recipient as a crucial variable in their primary fMRI analysis. This approach is logical, as the inequity between the two is a reliable indicator of the injustice they are examining. However, this inequity term is not included in equation (1). Even if this exclusion is the result of model selection, it is difficult to believe that this model accurately represents injustice. Pleaese clarify that inequity term has also an important role in modeling.

(b) It seems the authors employed RSS for model selection (equation (3)). As RSS doesn't compensate for the number of parameters, it can overfit data and select the model with the maximum number of parameters. The authors need to reanalyze the data using 10-fold cross-validation.

3. fMRI analysis

(a) The authors inconsistently switched their correction criteria in the fMRI analysis. They used a small volume correction for the amygdala but whole brain corrections for many cortical areas. A similar problem is apparent in ACC, PCC, and rTPJ, described on page 9. Even if f this is due to the authors hypothesizing the amygdala's involvement in inequity processing, previous studies have also reported that dorsal and ventral striatum and many cortical areas, including the insula, DLPFC, ACC, PCC, and IFG, are involved in the processing of inequity. Please apply correction criteria systematically for the fMRI analysis and also include these areas for consideration. 

(b) On page 9, the authors stated, "Interestingly, after correcting for multiple comparisons, we did not observe consistent differences in brain regions in subjective value representation between the stress and control groups." Why is this interesting rather than being not significant?

Reviewer #2: Thank you sending me this interesting manuscript. The manuscript investigated the effect of acute stress on third party punishment. The authors conclude that acute stress:

-decreased the third party's willingness to punish the violator

-decreased the severity of the punishment

-increased the willingness to help the victim

I think the findings are interesting and make a solid contribution to the field. I would be happy to read a revised version of the manuscript. Here are my comments and suggestions which I feel should be addressed before publication. 

Abstract

1. the use of the term "intervention severity bias" in the abstract is rather ambiguous. Either explain clearly what this is or remove it from the abstract

2. "Computational modeling revealed a shift in intervention severity bias from punishment toward help under stress. This finding is consistent with the increased dorsolateral prefrontal engagement observed with higher amygdala activity and greater connectivity with the ventromedial prefrontal cortex." Is this in the stress group? If so, please clarify.

Intro

3. When reviewing the literature on third party punishment under stress, the authors do not acknowledge an important recent meta-analysis which did not find consistent effects of acute stress on third party punishment:

-Does Stress Make Us More—or Less—Prosocial? A Systematic Review and Meta-Analysis of the Effects of Acute Stress on Prosocial Behaviours Using Economic Games: https://doi.org/10.1016/j.neubiorev.2022.104905

4. Additionally on p. 2 they write: 

"Acute stress can lead to increased activation of "theory of mind" 72 (ToM) brain regions (Tomova et al., 2017), which is fundamental for becoming more other-oriented with a preference for providing help (FeldmanHall et al., 2015)"

Again please see the meta-analysis above and also consider more recent findings which suggest that stress does not always lead to increased prosocial behaviour:

-Acute stress reduces effortful prosocial behaviour: https://doi.org/10.7554/eLife.87271.1

-Altruism under Stress: Cortisol Negatively Predicts Charitable Giving and Neural Value Representations Depending on Mentalizing Capacity: https://doi.org/10.1523/JNEUROSCI.1870-21.2022

5. p. 3 "we hypothesized that people under stress would prefer helping when third-party punishment and helping are in conflict because punishment is more executive control-dependent." Why should punishment be more dependent on executive control than helping?

There are many studies showing that prosocial behaviour requires cognitive control (e.g. inhibiting selfish responses). The dlPFC and parietal regions are consistently activated by prosocial tasks e.g. see this meta-analysis:

-Neural signatures of prosocial behaviors: https://doi.org/10.1016/j.neubiorev.2020.07.006

Why should this be different under stress?

Results 

4. The interaction between Group x Fairness x Intervention is not significant (p>.10). I would not label this as a 'trend effect'. Also, the decision to conduct the analysis only on the 'extreme' unfairness condition is not really justified. Why was this done?

Discussion

5. Could the findings be explained by differences in framing effects under acute stress (e.g. Pabst et al., 2013). Punishing always involved deducting money (loss frame) whereas being prosocial was always adding money (gain frame). How can the author be sure this is a social effect of acute stress rather than a framing effect?

6. p. 10 "a growing body of studies has demonstrated that stressed individuals have greater altruistic…" See comments/references in the introduction. The effects of acute stress on altruism/generosity are far from conclusive, please acknowledge this. 

7. p. 11 "helping behavior under stress is more intuitive and straightforward and becomes a more adaptive and habitual decision with relatively lower cognitive and computational expenditure." / "punishment is no longer an optimal strategy in dealing with other-regarding injustice events since it requires more cognitive and mentalizing resource involvement" (again see comment in the introduction). I feel the authors are just describing the results again but do not really explain why helping is less cognitively demanding under stress and/or why punishment is more cognitively demanding?

Reviewer #3: Summary of the paper:

The main questions of this paper are how acute stress changes the third-party's

---

## [Decision Letter · Decision Letter 2]

2 Feb 2024

Dear Dr Qin,

Thank you for your patience while we considered your revised manuscript "Acute stress reshapes third-party punishment and help decisions: Behavioral evidence and neurocomputational mechanisms" for publication as a Research Article at PLOS Biology. This revised version of your manuscript has been evaluated by the PLOS Biology editors, the Academic Editor and two of the original reviewers.

Based on the reviews and on our Academic Editor's assessment of your revision, we are likely to accept this manuscript for publication, provided you satisfactorily address the following data and other policy-related requests:

* We would like to suggest a different title to improve readability: "Acute stress when witnessing injustice increases the willingness of bystanders to help the victim instead of punishing the perpetrator"

* Please add the links to the funding agencies in the Financial Disclosure statement in the manuscript details.

* Please include information about the study has been conducted according to the principles expressed in the Declaration of Helsinki.

DATA POLICY:

Regardless of the method selected, please ensure that you provide the individual numerical values that underlie the summary data displayed in the following figure panels as they are essential for readers to assess your analysis and to reproduce it: Figure 1B, 1C, 1D, all panels of Figure 2, 3B, 3D, 3F, 3G, and similar panels in the supplementary figures

CODE POLICY

Per journal policy, as the code that you have generated is important to support the conclusions of your manuscript, we require that you make it available without restrictions upon publication. Please ensure that the code is sufficiently well documented and reusable, and that your Data Statement in the Editorial Manager submission system accurately describes where your code can be found.

We expect to receive your revised manuscript within two weeks. 

*Published Peer Review History*

*Press*

Sincerely,

Christian

Christian Schnell, PhD

Senior Editor

cschnell@plos.org

PLOS Biology

Reviewer remarks:

Reviewer #1: Thank you for your intensive revision of the manuscript.

The revised version is much clearer and convincing, which successfully addressed all of my concers.

In addition, I now understand the computational model and model selection part properly. Thank you for the explanations.

Reviewer #2: I am satisfied that the authors have addressed my concerns.

---

## [Editor Report · Decision Letter 3]

9 Feb 2024

Dear Dr Qin,

Thank you for your patience while we considered your revised manuscript "Acute stress reshapes third-party punishment and help decisions: Behavioral evidence and neurocomputational mechanisms" for publication as a Research Article at PLOS Biology. 

Thank you for addressing the editorial requests and for sending alternative title suggestions. I've taken your suggestions into account and hope that the title as below is acceptable to you. If so, could you please change the title accordingly in your manuscript files and in Editorial Manager?

* New title: "Acute stress during witnessing injustice shifts third-party interventions from punishing the perpetrator to helping the victim"

* The information on whether your study adhered to the guidelines set out in the declaration of Helsinki is still absent from the manuscript. For example "The protocol was designed and performed according to the principles of the Helsinki Declaration and approved the Institutional Review Board of Department of Psychology at Renmin University of China (IRB2017052701). All participants signed written informed consent. "

* And finally, can you please add a statement to the corresponding figure legends where the raw data can be found? For example: "Source data can be found at https://osf.io/fkae9/

We expect to receive your revised manuscript within two weeks. 

*Published Peer Review History*

*Press*

Sincerely,

Christian

Christian Schnell, PhD

Senior Editor

cschnell@plos.org

PLOS Biology

---

## [Editor Report · Decision Letter 4]

20 Feb 2024

Dear Dr Qin,

Thank you for the submission of your revised Research Article "Acute stress during witnessing injustice shifts third-party interventions from punishing the perpetrator to helping the victim" for publication in PLOS Biology. On behalf of my colleagues and the Academic Editor, Matthew Rushworth, I am pleased to say that we can in principle accept your manuscript for publication, provided you address any remaining formatting and reporting issues. These will be detailed in an email you should receive within 2-3 business days from our colleagues in the journal operations team; no action is required from you until then. Please note that we will not be able to formally accept your manuscript and schedule it for publication until you have completed any requested changes.

PRESS

Sincerely, 

Christian

Christian Schnell, PhD

Senior Editor

PLOS Biology

cschnell@plos.org